# Improved prediction of solvation free energies by machine-learning polarizable continuum solvation model

Amin Alibakhshi [1✉] & Bernd Hartke [1]

Theoretical estimation of solvation free energy by continuum solvation models, as a standard approach in computational chemistry, is extensively applied by a broad range of scientific disciplines. Nevertheless, the current widely accepted solvation models are either inaccurate in reproducing experimentally determined solvation free energies or require a number of macroscopic observables which are not always readily available. In the present study, we develop and introduce the Machine-Learning Polarizable Continuum solvation Model (ML-PCM) for a substantial improvement of the predictability of solvation free energy. The performance and reliability of the developed models are validated through a rigorous and demanding validation procedure. The ML-PCM models developed in the present study improve the accuracy of widely accepted continuum solvation models by almost one order of magnitude with almost no additional computational costs. A freely available software is developed and provided for a straightforward implementation of the new approach.

[1] Theoretical Chemistry, Institute for Physical Chemistry, Christian-Albrechts-University, Olshausenstr. 40, Kiel, Germany. ✉email: alibakhshi@pctc.uni-kiel.de

Free energy of solvation is one of the key thermophysical properties in studying thermochemistry in solution, where the majority of real-life chemistry happens. In theoretical studies of solution chemistry, estimation of free energies allows evaluation of reaction rates and equilibrium constants of physical or chemical reactions of interest. Nevertheless, direct evaluation of free energies in solution can be quite challenging since it sometimes requires appropriate sampling of phase space[1–3] and appropriate treatment of the non-covalent interactions between the solvent and solute, which can have a remarkable impact on electronic structures of both the solvent and solute and consequently on the microscopic and macroscopic observables[4,5].

Theoretical approaches for evaluating physical chemistry behind solvation free energy can be generally divided into two main categories, namely explicit solvent and implicit solvent approaches. In explicit solvent approaches, solvent molecules are treated explicitly, and the free energy is typically evaluated by analyzing the trajectory of time evolution of phase space obtained via molecular dynamics or Monte Carlo simulations. For that end, a number of efficient free energy estimators have been developed in the past decades such as thermodynamic integration, free-energy perturbation, and histogram analysis methods[6].

Despite obvious advantages of applying the explicit solvent methods such as retaining the physically proper picture of discrete solvent molecules, they suffer by a number of limitations when applied to free-energy estimation. For example, in case of applying methods which evaluate the free energy through alchemical transformations (e.g., thermodynamic integration or free energy perturbation), defining intermediate states and pathways between the endpoints appropriately can be quite tricky[7]. Also, necessity of employing appropriate force fields, which for many solute-solvent mixtures requires to develop or reparametrize a force field, and running the simulations and trajectory analyses can be laborious and time-taking tasks.

To overcome the mentioned limitations, the implicit solvent approach has been developed and is widely applied as standard method for studying solvent effects in computational chemistry. In implicit solvent approaches, the solvent molecules are treated implicitly as a continuous medium and the solute is placed in a cavity of this implicitly defined solvent. The solute-solvent interactions are then evaluated via considering the solvent polarization due to the solute charge distribution and its resulting potential field acting on the solute, known as the reaction field[5]. For a moderate level of theory and medium-sized molecules, implicit solvent approaches can yield a reasonable estimation of the solvation free energy in few seconds to few minutes on a normal desktop PC, while for explicit solvent approaches it might take from hours to days.

The most widely applied implicit solvent approaches are those based on the so-called polarizable continuum model (PCM) proposed by Tomasi and co-workers[8]. In polarizable continuum models, the solvation free energy is constructed by summing the contributions of electrostatic interactions including electronic, nuclear, and polarization interactions ($\Delta G_{ENP}$), changes in free energy by solvent cavity formation, dispersion energy and local solvent structure changes ($G_{CDS}$), and corrections for differences in molar densities in the two phases compared with the standard state ($\Delta G°_{cons}$). The contributions of electrostatic interactions are evaluated by iteratively solving the following relationship:

$$\Delta G_{ENP} = \langle \Psi^{(1)}|H + \frac{1}{2}V|\Psi^{(1)}\rangle - \langle \Psi^{(0)}|H|\Psi^{(0)}\rangle \qquad (1)$$

which is known as the self-consistent reaction-field (SCRF) calculations[5]. Here, superscripts (0) and (1) refer to the gas and solution phases, respectively, and $V$ is the potential energy operator resulting from the reaction field. Various constructions

of the potential energy operator as well as $G_{CDS}$ have resulted in different continuum solvation models. The parallel existence of several continuum solvation models is a good indicator that each of them has its own strengths and weaknesses, and choosing a single, optimal model is not trivial. It is totally impossible to provide a detailed overview here; a 2005 review of implicit solvation models[9] covered 95 pages and cited 936 references. In the present study, we only consider the most widely used PCM-based models.

One of simplest and yet successful continuum solvation models is CPCM which implements the conductor-like screening solvation boundary condition within the PCM framework. In CPCM, the following correction of the polarization charge densities by the scaling factor x is employed[10]:

$$f(\varepsilon) = \frac{\varepsilon - 1}{\varepsilon + x} \qquad (2)$$

where $\varepsilon$ is the solvent dielectric constant. One main advantage of CPCM is its much simpler defined boundary conditions. More importantly, unlike more advanced PCM-based models which require the normal component of the solute electric field as input, CPCM only requires the solute electrostatic potential; for this reason it is much less affected by outlying charge errors (OCE)[11,12]. A more versatile model exploiting the conductor-like screening solvation boundary condition is COSMO-RS, developed by Klamt and co-workers[13,14], which although initially proposed in 1995, still is one of the most accurate available continuum solvation models. A more sophisticated treatment of the boundary condition is implemented in the integral equation formalism of PCM (IEF-PCM) taking into account apparent surface charge isotropic[15] or anisotropic[16] dielectric continuum solvation. Another extensively used continuum solvation model is the SMx family of methods which specifically focuses on more accurate estimation of the solvation free energy[4,5].

We already discussed the main advantages of continuum solvation models such as their efficiency in terms of computational cost. Nevertheless, it should be noted that all this has become possible for a considerable amount of assumptions and simplifications on the physics of the problem, such as overlooking the conformational entropy of solvent and solute which can have a significant contribution on the total free energy[17], neglecting the site-specific solute-solvent interactions and decoupling the polar and nonpolar components of free energies and considering them independent, linear and additive[18,19]. The inaccuracies resulting from such simplifications are commonly compensated for via incorporating additional macroscopic observables as well as adjustable parameters in the solvation models. In the CPCM model for example, this is achieved by implementing an ad hoc modification of the atomic radii via defining a number of adjustable parameters and empirical descriptors, such as the number of bonded hydrogens and the number of bonded active atoms[10]. In the COSMO-RS model, it is achieved by ad hoc modification of the interaction energies and effective contact area via some adjustable parameters[14].

In contrast, in the SMx family of methods, to provide a more accurate estimation of the solvation free energy, an ad hoc modification of the $G_{CDS}$ term in (1) has been proposed. For that end, employing additional macroscopic observables in the model has been considered[4], including the refractive index, Abraham's hydrogen bond acidity and basicity of the solute, macroscopic surface tension of the solvent at the air/solvent interface at 298.15 K, the square of the fraction of solvent atoms that are aromatic carbon atoms, and the square of the fraction of solvent atoms that are F, Cl, or Br. Although these employed macroscopic observables indirectly introduce more physics into the model and hence provide the chance to make predictions of solvation free

energies more universal, except for the last two they are not readily available for many new compounds and their experimental or theoretical evaluation is not straightforward.

In a number of recent studies, Machine Learning (ML) has been exploited to map the highly complicated relationship between solvation free energy and potentially relevant macroscopic or microscopic observables.

Wang et al. employed a pool of 30 molecular representations which all are either per atom reaction field energies or partial charges, as the input of the learning-to rank (LTR) machine learning algorithm, resulting in a root mean squared error (RMSE) of 1.05 kcal/mol[18]. Borhani et al. developed a QSPR model which requires 12 experimentally determined properties of solvent and 9 QM derived representations of solute as model input, yielding a Mean Unsigned Error (MUE) of 0.43 kcal/mol[20]. Hutchinson and Kobayashi proposed a structure property relationship for prediction of hydration free energy which yields a RMSE of 1.65 kcal/mol[21].

Another recent example is the kernel-based machine learning model of Rauer and Bereau which is developed to predict the free energy of solvating small organic molecules containing C, H, O, and N atoms in pure water via implicit-solvent molecular dynamics simulations[22]. For a 39-parameter model they reported a MUE of 1.06 kcal/mol.

The most recent example of employing machine learning for prediction of solvation free energy is the model developed by Vermeire and Green[23]. Their model is developed based on the transfer of knowledge learned through one million data of QM evaluated free energies and fine tuning it to accurately reproduce the experimentally determined solvation free energies. They reported a MUE of 0.21 kcal/mol for their model which is currently the most accurate ever reported result for prediction of solvation free energy.

In the present study, we propose a machine-learning-based PCM model, which, similar to other conventional continuum solvation models, is based on considering the solvent as a continuous medium and calculating the solvation energy components of a solute placed in the cavity of this medium by the SCRF procedure. Nevertheless, unlike the conventional PCM models which propose simple and ad hoc expressions to integrate and modify those calculated energy components, we employ machine learning for this purpose and show its efficiency in substantial improvements of the predictability of solvation free energy.

## Results and discussions

After setting up and training the neural networks and screening the appropriately trained models via the post-validation strategy discussed in the previous section, the best results with MUE of 0.52526 and 0.40011 kcal/mol were observed for the computations at B3LYP/6–31 G* and DSD-PBEP86-D3/def2TZVP levels of theory, respectively. The two models employed SCRF energy components and solvation free energy computed via $CPCM_{x=0.5}$ solvation model in both cases and 100 and 130 hidden layer neurons, respectively. These two models are denoted by ML-PCM(B3LYP) and ML-PCM (DSD-PBEP86) hereafter, respectively. Details of the selected input variables and implementation instructions for all selected models are provided in Supplementary Software 1. These results show a substantial improvement compared to the original continuum solvation model $CPCM_{x=0.5}$, which for the same dataset yielded MUE of 3.1611 and 2.9130 kcal/mol, respectively.

In comparison to the SMD model, which for the same dataset and solvation free energy computations at B3LYP/6–31 G* and DSD-PBEP86-D3/def2TZVP levels yields MUE of 0.78623 and 0.85396 kcal/mol, respectively, the obtained results still show a higher accuracy, without requiring additional solvent parameters

needed in the SMD approach. In comparison to the MUE of 0.4214 kcal/mol reported by Klamt and Diedenhofen[24] for employing one of the recent versions of the COSMO-RS model for the same dataset, the ML-PCM(DSD-PBEP86) provides a slightly higher accuracy. Also, in terms of maximum unsigned error, the two ML-PCM models which yield maximum unsigned error of 6.2252and 3.8799 kcal/mol, respectively, are more accurate than that of COSMO-RS for which this value is 6.8701 kcal/mol. For other continuum solvation models studied for the same dataset, the maximum unsigned error of the SMD, PCM, CPCM and $CPCM_{x=0.5}$ were 11.311, 12.75, 12.2, 12.6 kcal/mol for B3LYP/ 6–31 G* and 11.311, 12.83, 12.31, 12.68 kcal/mol for DSD-PBEP86-D3/def2TZVP levels of theory, which are all substantially higher than those achievable by the ML based models.

The higher accuracy of the predicted solvation free energies by the COSMO-RS model compared to the other conventional solvation models also motivated us to study neural networks which take SCRF energy components computed via PCM or CPCM models in addition to the solvation free energies predicted via COSMO-RS as neural network feeds. For these updates, the best results with MUEs of 0.26057 and 0.24387 kcal/mol and maximum unsigned errors of 7.1349 and 2.9154 kcal/mol were obtained for energy components calculated via $CPCM_{x=0.5}$ and CPCM solvation models, 130 and 120 hidden layer neurons, and computations at B3LYP/6–31 G* and DSD-PBEP86-D3/ def2TZVP levels of theory, respectively. These two models, which are denoted by ML-PCM/COSMO-RS(B3LYP) and ML-PCM/ COSMO-RS(DSD-PBEP86) hereafter, respectively, show a remarkable improvement in predicted solvation free energy compared to those obtained via the original implementation of COSMO-RS reported by Klamt and Diedenhofen[24]. This implies considerable flexibility of the proposed approach in improving accuracy of various solvation models. Nevertheless, it should be noted that the solvation free energies evaluated by COSMO-RS which were used as additional model inputs in the present study were evaluated using the 2015 version of that method. Using free energies evaluated by more recent versions of COSMO-RS and also the energy terms computed with this method, will probably result in more accurate predictions of the solvation free energy by the presented ML-PCM.

As the most important parameter in developing ANN models, we studied the impact of the selected number of hidden layer neurons on the performance of the developed machine learning models. As can be seen in Fig. 1, by increasing the number of hidden layer neurons, the predictability of the solvation free

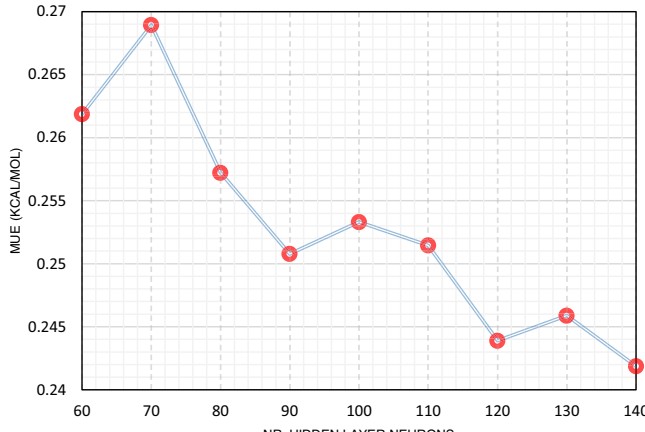

**Fig. 1 MUE of developed ML-PCM/COSMO-RS(B3LYP) models versus the number of hidden layer neurons.** The general trend shows the reducing pattern in MUE with increasing the size of the neural network model.

energy is generally improved. This is due to the larger number of adjustable parameters of the resulting models and their consequently higher flexibility to map complicated functionalities. However, at the same time this may reduce the extrapolation capability of the model, i.e., it may reduce performance when applied to samples remarkably different from those already examined in developing the models.

To investigate the impact of the number of hidden layer neurons on extrapolation performance of the models developed in the present study, we re-examined the trained models for out-of-sample predictions, following the approach proposed by Vermeire and Green[23]. For that end, we compared the results of models for which a group of samples with either a specific element or a specific solvent were included in the training dataset with the same models trained with a dataset excluding that specific group of samples. We studied out-of-sample prediction performance for 20 solvents and 6 solute elements most frequently encountered in our studied dataset. The obtained results are reported in Tables 1 and 2. According to the results, the developed models show an excellent extrapolation capability for out of sample predictions of solvent splits, while for the element splits, the extrapolation is slightly less accurate. Furthermore, except for the element Br, the out-of-sample predictions tested for ML-PCM/COSMO-RS(B3LYP) are within chemical accuracy.

A comparison of predicted and experimentally determined free energies is depicted in Fig. 2. As can be seen, the linear correlation between the predicted and reference data is more evident for the newly derived models, compared to the conventionally accepted ones.

The overall results obtained via newly developed ML models are compared with various other models proposed in the literature in Table 3. Although a more informative comparison would be possible if different models were compared for the same dataset and, if applicable, the same level of theory, the larger size of the benchmark dataset used in the present study compared to most of the other works confirms the superior accuracy of the newly proposed method compared to the majority of the widely accepted ones. In comparison to the model developed by Vermeire and Green[23] which yields MUE of 0.21 kcal/mol, our results are slightly less accurate, but it should be noted that our results are obtained for a much lower number of neurons and model parameters.

Furthermore, it should be noted that the inaccuracies inherent in the reference data of solvation free energies (Aleatoric uncertainty) can also impact both the training efficiency and inferences about model performances, as pointed out by Vermeire and Green[23].

To summarize, we have demonstrated substantial improvements of continuum solvation models in evaluating solvation free energy with the help of machine learning. For that end, we proposed a more versatile machine learning assisted integration of the continuum solvation energy components calculated in SCRF computations which can be used to modify the predicted solvation free energy by various solvation models. It allowed us to achieve

**Table 1 Out-of-sample predictions for solvent splits.**

| Solvent | Nr. Samples | ML-PCM/COSMO-RS(B3LYP) | | ML-PCM/COSMO-RS(DSD-PBEP86) | |
| --- | --- | --- | --- | --- | --- |
| | | MUE (solvent included) | MUE (solvent excluded) | MUE (solvent included) | MUE (solvent excluded) |
| Water | 261 | 0.13921 | 0.53856 | 0.12724 | 0.52107 |
| n-Octanol | 199 | 0.21116 | 0.40528 | 0.19416 | 0.34079 |
| n-Hexadecane | 184 | 0.47931 | 0.63312 | 0.42914 | 0.38652 |
| Chloroform | 102 | 0.2962 | 0.33126 | 0.27975 | 0.28894 |
| CycloHexane | 88 | 0.27941 | 0.30729 | 0.30521 | 0.35877 |
| CarbonTetraChloride | 73 | 0.37704 | 0.38958 | 0.30407 | 0.31146 |
| Benzene | 71 | 0.21953 | 0.24581 | 0.37323 | 0.52627 |
| DiethylEther | 66 | 0.23975 | 0.29187 | 0.22181 | 0.22156 |
| Heptane | 64 | 0.41215 | 0.4795 | 0.2033 | 0.19233 |
| n-Hexane | 57 | 0.19548 | 0.19648 | 0.28332 | 0.3775 |
| Toluene | 49 | 0.22219 | 0.20023 | 0.31435 | 0.33986 |
| Xylene-mixture | 46 | 0.25694 | 0.22309 | 0.27209 | 0.27789 |
| DiChloroEthane | 37 | 0.38469 | 0.49085 | 0.22075 | 0.28748 |
| n-Decane | 37 | 0.21171 | 0.25761 | 0.15559 | 0.17148 |
| ChloroBenzene | 36 | 0.2183 | 0.2374 | 0.20119 | 0.22425 |
| n-Octane | 35 | 0.13265 | 0.13455 | 0.17431 | 0.19447 |
| 2,2,4-TriMethylPentane | 32 | 0.2097 | 0.20388 | 0.23426 | 0.23618 |
| EthylBenzene | 27 | 0.20878 | 0.23166 | 0.20471 | 0.24331 |
| BromoBenzene | 24 | 0.16054 | 0.22188 | 0.13648 | 0.18478 |
| Decalin-mixture | 24 | 0.39408 | 0.44484 | 0.31164 | 0.33004 |

**Table 2 Out-of-sample predictions for element splits.**

| Element | Nr. Samples | ML-PCM/COSMO-RS(B3LYP) | | ML-PCM/COSMO-RS(DSD-PBEP86) | |
| --- | --- | --- | --- | --- | --- |
| | | MUE (element included) | MUE (element excluded) | MUE (element included) | MUE (element excluded) |
| N | 611 | 0.25549 | 0.40139 | 0.25486 | 0.37139 |
| F | 81 | 0.29188 | 0.38812 | 0.32345 | 0.48087 |
| P | 62 | 0.12927 | 0.64773 | 0.20648 | 0.95936 |
| S | 91 | 0.26592 | 0.50868 | 0.29104 | 0.53079 |
| Cl | 174 | 0.25295 | 0.5194 | 0.17956 | 0.47383 |
| Br | 102 | 0.25005 | 1.4559 | 0.26268 | 0.91972 |

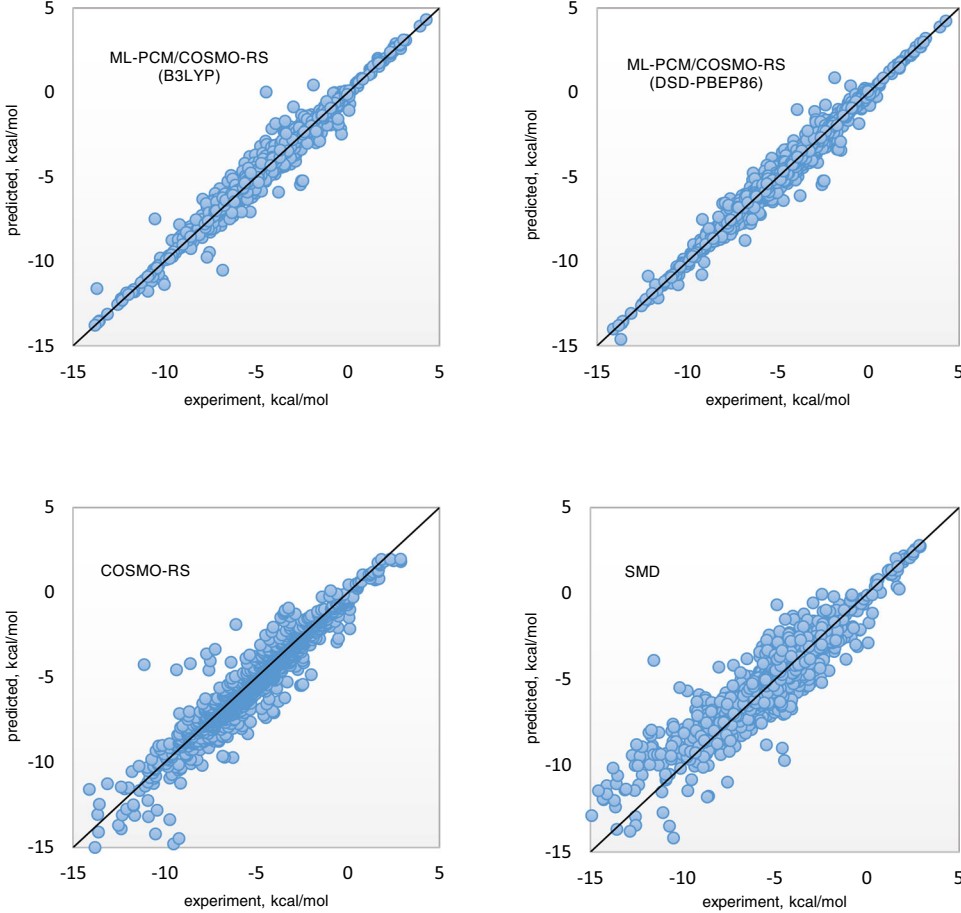

**Fig. 2 Comparison of experimentally determined and predicted solvation free energies for various solvation models.** The results show a higher correlation between the experimentally determined and predicted data for the proposed machine learning solvation models compared to the SMD or COSMO-RS models.

accurate predictions of solvation free energy with MUE as low as 0.2439 kcal/mol for a large dataset of 2493 binary mixtures of 435 neutral solutes and 91 solvents from diverse chemical families.

## Methods

**Dataset.** To benchmark our results, we used the solvation free energy data of 2493 binary mixtures of 435 neutral solutes and 91 solvents from diverse chemical families available in the Minnesota solvation database[4]. The full list of the studied samples can be found as Supplementary Data 1.

**Computational details.** The performance of models is reported as mean unsigned error (MUE) and root mean squared error (RMSE) defined as:

$$MUE = \frac{1}{N}\Sigma\left(\left|y_i^{\exp} - y_i^{pred}\right|\right) \quad (3)$$

$$RMSE = \left(\frac{1}{N}\Sigma\left(\left(y_i^{\exp} - y_i^{pred}\right)^2\right)\right)^{\frac{1}{2}} \quad (4)$$

where $y_i^{\exp}$ and $y_i^{pred}$ are experimentally determined and predicted solvation free energies, respectively.

Prior to SCRF computations, all solute geometries were optimized in vacuo at the B3LYP/6-31 G*level of theory. Using the optimized structures, the SCRF principal energy components listed in Table 4 were computed for each compound at the B3LYP/6-31 G* and DSD-PBEP86-D3/def2TZVP levels of theory. The latter method as a double hybrid has been shown to yield more precise charge distributions and energy estimations compared to lower-rung DFT or MP2 methods, for a cost comparable to that of the MP2 calculation[25].

The SCRF energy components listed in Table 4 were computed for two widely accepted polarizable continuum models, namely the IEF-PCM and CPCM, as implemented in Gaussian 16 (ref. [26]). For CPCM, the default value of zero is considered as the scaling factor $x$ in relationship (2). However, a value of 0.5 has been shown to be a more reasonable choice for this scaling factor[11,27]. Therefore, in

addition to the default implementation of CPCM in Gaussian 16, we also employed a CPCM model with a scaling factor of $x=0.5$ and denote it by CPCM$_{x=0.5}$. For that, we replaced the original dielectric constant of the solvent with an effective dielectric constant $\widetilde{\varepsilon}(\varepsilon, x)$ calculated via:

$$\widetilde{\varepsilon}(\varepsilon, x) = \frac{\varepsilon + x}{x + 1} \quad (5)$$

as suggested by Klamt et al.[11]. For comparison purposes, we also calculated the solvation free energy via the SMD approach.

We employed feed-forward neural networks to map the relationship between the solvation free energy and the calculated SCRF energy components, which in addition to the solvation free energy estimated by the applied continuum solvation model and to the dielectric constant of the solvent, comprised our model inputs.

The obtained pool of model inputs was further screened using the Minimum Redundancy and Maximum Relevance (MRMR) algorithm[28] resulting in various 8–16 membered combinations of those variables. MRMR is a highly efficient algorithms for selecting most effective sets of variables for developing robust machine-learning-based models[29]. For each number of selected variables, 25 different settings of the MRMR algorithm were applied, distinguished by the employed quantization level, level of dependency, forward or backward variable selection and considering pseudo-samples based on Bayesian statistics or not[28]. In many cases, this resulted in diversely selected set of variables, even for the same applied level of theory and continuum solvation model.

In the next step, various configurations of neural network models were set up and their reliability were examined with a demanding procedure based on the guidelines presented in a previous study[30]. Accordingly, we assigned large parts of the dataset for test (25%) and validation (15%), and only 60% of the dataset compounds were used for training the models.

To improve the transferability of the developed models for out-of-sample predictions, validation and test sets were selected in a way to include either solvent or solute elements not available in the training set.

We employed Levenberg-Marquardt backpropagation and Gradient descent backpropagation training algorithms, and hidden layer transfer functions of the logarithm-sigmoid and tangent-sigmoid types[31]. We only employed neural

**Table 3 Comparison of the results of the new method with other models.**

| Method | Source | Nr. Samples | Nr. Solvents | Nr. Solutes | Deviation measure | Deviation (kcal/mol) |
|---|---|---|---|---|---|---|
| ML-PCM/COSMO-RS(DSD-PBEP86) | Present study | 2224 | 88 | 300 | MUE | 0.24387 |
| | | | | | RMSE | 0.37252 |
| ML-PCM/COSMO-RS(B3LYP) | Present study | 2224 | 88 | 300 | MUE | 0.26057 |
| | | | | | RMSE | 0.43623 |
| ML-PCM (DSD-PBEP86) | Present study | 2488 | 91 | 435 | MUE | 0.40011 |
| | | | | | RMSE | 0.56014 |
| ML-PCM (B3LYP) | Present study | 2493 | 91 | 435 | MUE | 0.52526 |
| | | | | | RMSE | 0.75112 |
| Other models found in the literature: Machine learning | Vermeire and Green[23] | 10145 | 291 | 1368 | MUE | 0.21 |
| | | | | | RMSE | 0.44 |
| COSMO-RS | Klamt and Diedenhofen[24] | 2346 | 91 | 318 | MUE | 0.42145 |
| | | | | | RMSE | 0.69644 |
| SM12 | Marenich et al.[32] | 2403 | 91 | 352 | MUE | 0.5457-0.6717 |
| QSPR | Borhani et al.[20] | 1777 | 210 | 295 | MUE | 0.43 |
| | | | | | RMSE | 0.52 |
| DCOSMO-RS | Klamt and Diedenhofen[24] | 2346 | 91 | 318 | MUE | 0.6584 |
| | | | | | RMSE | 0.99724 |
| SMD (B3LYP) | Present study | 2493 | 91 | 435 | MUE | 0.78623 |
| | | | | | RMSE | 1.1633 |
| SMD (DSD-PBEP86) | Present study | 2488 | 91 | 435 | MUE | 0.85396 |
| | | | | | RMSE | 1.3362 |
| Feature Functional Theory | Wang et al.[18] | 668 | 1 (water) | 668 | RMSE | 1.05 |
| kernel-based machine learning | Rauer and Bereau[22] | 355 | 1 (water) | 355 | MUE | 1.06 |
| atoms-in-molecules neural network | Zubatyuk et al.[33] | – | – | 414 | MUE | 1.1 |
| Structure-Property Relationship | Hutchinson and Kobayashi[21] | – | 1 (water) | – | RMSE | 1.65 |
| CPCM(B3LYP) | Present study | 2493 | 91 | 435 | MUE | 2.6942 |
| | | | | | RMSE | 3.1733 |
| PCM(B3LYP) | Present study | 2493 | 91 | 435 | MUE | 2.9054 |
| | | | | | RMSE | 3.3948 |
| CPCM$_{x=0.5}$(B3LYP) | Present study | 2493 | 91 | 435 | MUE | 2.9130 |
| | | | | | RMSE | 3.3985 |
| CPCM(DSD-PBEP86) | Present study | 2488 | 91 | 435 | MUE | 2.9651 |
| | | | | | RMSE | 3.4426 |
| PCM (DSD-PBEP86) | Present study | 2488 | 91 | 435 | MUE | 3.1569 |
| | | | | | RMSE | 3.6445 |
| CPCM$_{x=0.5}$(DSD-PBEP86) | Present study | 2488 | 91 | 435 | MUE | 3.1611 |
| | | | | | RMSE | 3.6466 |

**Table 4 The components of the continuum solvation model.**

1. Solvation free energy calculated by the continuum solvation model
2. $\langle \Psi^{(0)} | H | \Psi^{(0)} \rangle$
3. $\langle \Psi^{(0)} | H + V^{(0)}/2 | \Psi^{(0)} \rangle$
4. $\langle \Psi^{(0)} | H + V^{(1)}/2 | \Psi^{(0)} \rangle$
5. $\langle \Psi^{(1)} | H | \Psi^{(1)} \rangle$
6. $\langle \Psi^{(1)} | H + V^{(1)}/2 | \Psi^{(1)} \rangle$
7. Interaction energy of unpolarized solute and polarized solvent
8. Interaction energy of polarized solute and polarized solvent
9. Solute polarization energy
10. Total electrostatic interaction energy
11. Cavity surface area
12. Cavity volume
13. Total kinetic energy
14. Total potential energy
15. Sum of kinetic and potential energy

networks with one hidden layer and 1 to 140 neurons in the hidden layer, with intervals of 10 neurons for ANNs with more than 50 neurons in the hidden layer. For each neural network configuration, training was carried out for 60 randomly selected training, validation and test sets, and for each one 40 different initializations of weight and bias constants of the neural networks were made. Above all, to avoid getting misleading data affected by favorable or unfavorable division of dataset into training, validation and test sets, the post validation strategy proposed in a previous study[30] was carried out. Accordingly, during the initial training of the neural networks, for the models which yielded mean absolute percentage errors lower than 22%, the final optimized weights and bias constants of the neural network models were recorded. These recorded constants were used as the initial guess to train, validate and test the same neural network configurations but under 100 different randomly selected training, validation and test sets. The models for which in at least 80 out of 100 iterations their test and training sets

errors had the same means and variances as evaluated by the two sample t-test method with 5% significance level were considered as reliably trained models. For them, the average of the ANN-predicted results in all repeats were reported as the performance of that model. Setting up and running the neural network models were implemented in Matlab software. A freely available C++ code for practical use of our proposed ML-PCM models, with detailed user instructions, is provided in Supplementary Software 1.

All the computations were carried out on the High Performance Computing center clusters of the Christian-Albrechts-University of Kiel.

## Data availability

All data produced in this study are available and can be provided by contacting the corresponding author.

## Code availability

The source file of the C++ code developed for implementing the proposed method with detailed used instructions are available in Supplementary Software 1 or can be provided by contacting the corresponding author.

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

## Acknowledgements

The authors wish to thank Karsten Balzer at the high performance computing center of Kiel University for his support and assistance in running the computations there. The Authors wish to thank the referees for their careful review of our work and fruitful discussions and comments.

## Author contributions

A.A. has contributed to method development, carried out the computations and contributed to writing the manuscript. B.H. supervised the project and contributed to method development and writing the manuscript.

## Funding

## Competing interests

The authors declare no competing interests.
