## [Peer Review File · Nature Communications]

Reviewer #1 (Remarks to the Author):

The authors develop a machine learning based continuum solvation model with an aim to improve the predictability of solvation free energy. The performance and reliability of the model are tested in comparison with other currently available models on the market.

As a manuscript submitted to *Nature Communications*, I had expected a more advanced implicit solvation model augmented with some machine-learning-based technique or an SCRF-applicable QM machine learning model, for example, <https://doi.org/10.1126/sciadv.aav6490>, while I was reviewing the manuscript. However, I found that the proposed model is just a neural-network-based SPR model that uses several QM results and empirical properties for molecular description, similar to other published models. Nevertheless, the authors claim that their proposed model is ML-PCM. Although the proposed model showed reasonable results, I do not see neither any novelty nor physical insight of the manuscript considering other published works, for example, <https://doi.org/10.1039/C9SC02452B>, <https://doi.org/10.1039/C8CP07562J>, <https://doi.org/10.1063/5.0012230>, <https://doi.org/10.1021/acs.jcim.0c00600>. Specific comments are as follow:

1. The authors mainly compare their results with three implicit solvation models: SMD, IEF-PCM, and C-PCM. There are more advanced and accurate models like COSMO-RS or SM12, and they also offer published results for the same database: <https://doi.org/10.1021/ct300900e>, <https://doi.org/10.1021/jp511158y>.
2. Since the training task uses relatively small data points, the authors need to discuss the model's robustness to overfitting.
3. The proposed model uses QM-calculation results as molecular descriptors. At least the authors must account for the proposed model has distinct advantages than other SPR models, which do not require such expensive calculations for description.

Overall, I do not recommend the publication of the current manuscript in *Nature Communications*.

Reviewer #2 (Remarks to the Author):

Comments to "Improved prediction of solvation free energies by Machine-Learning Polarizable Continuum solvation Model"

The manuscript proposes a machine learning model to improve results from PCM calculations using quantum molecular descriptors and additional macroscopic properties. The methodology clearly improves the results obtained by PCM calculations to chemical accuracy required for modeling on a random selection of experimental data. Even though the results look good, I have some concerns related to necessity of available macroscopic properties as input to the machine learning model, as well as the application range of the model. Also, there is a widely used commercially-available program COSMO-RS / COSMOtherm developed many years ago which is similar (starts from a type of PCM calculation, then adds a few corrections to achieve higher accuracy), but the authors apparently haven't compared their method to COSMO. To this reviewer, it is not clear how the new method is novel compared to COSMO-RS/COSMOtherm, which appears to be essentially the same concept. More important than the novelty, the authors need to show that their empirical correction to PCM is significantly more accurate or more convenient than COSMO. (I think in reality COSMO is more convenient since it doesn't require some inputs, e.g. the normal boiling point of the solute, which is often unavailable.) For these reasons I do not recommend publication of this manuscript in present form, and also if it is published I think it should not be a Communication but an ordinary article. Detailed comments are given below.

Major Comments:

1. Introduction

- The authors claim that one of the advantages of their approach is the strong reduction in the need for macroscopic properties that are usually not readily available. The introduction should highlight more what these properties are and how the properties used in this work are more easily accessible. Some of the macroscopic properties required by the current method seem quite difficult to obtain in many practical situations.

- With regard to importance to this work, I believe that there are 3 important topics missing in the introduction:

1. The state of the art COSMO-RS method developed by Klamt and implemented in COSMOtherm. This method typically outperforms other PCM, such as SMD. Since this software is licensed, we understand that this method is not used in the present work and the authors focus on other PCM methods for comparison, but it should at least be discussed in the introduction. We note that the authors apparently were willing to use (and implicitly to require anyone using their method to also use) the commercial GAUSSIAN software, which has a very restrictive license. Why were they unwilling to use the COSMOtherm software which is so relevant to their research? Also a comparison of the uncertainty of this method to the new ML approaches would help the manuscript to compare to the state of the art.

2. The uncertainty of each of the PCM models used should be discussed (apart from the results in Table 2). How do the selected PCM models compare in general to other PCM models, do they perform better on a separate test set and is this the reason they are selected? Do they require fewer macroscopic properties compared to other methods and which macroscopic properties are required?

3. What is the applicability range for the used PCM models? Typically these models are parameterized to perform well within a certain subset of chemical space. i.e. what is the Scope of the new method proposed in this paper?

2. Computational details

- The geometry for the solutes is optimized in vacuo and that geometry is accounted for during computations with the implicit solvent model approach. Was there an attempt to ensure that you found the lowest-energy conformer? How do the authors account for the change in geometry of the lowest energy conformer in different solvents. Why are the geometries not optimized while

accounting for the implicit solvent? Lowest energy conformers will be different in polar and apolar solvents.

- The authors agree that the MUPE might be a more appropriate metric to define the error for solvation free energies compared to MUE. However, many of the solvation free energies have values close to 0.0 kcal/mol, which can complicate the use of MUPE and result in near-infinite values. Although it is not common, we think it is possible that for some solute-solvent pairs the solvation free energy could be negative which would cause a problem with the MUPE formula. How do the authors handle this?

- Can the authors give an indication of the computational time for each of the different methods used?

- The authors use 9 macroscopic properties to optimize the neural network. At least two of those, the normal boiling point of the solute and the dielectric constant of the solvent, are very important inputs to the neural network. The authors claim that these are readily available, which is the case for common small-molecule solvents and solutes. However, when accounting for new molecules these properties are more difficult to predict and the model heavily relies on their availability. Also many larger solutes of great interest (e.g. known or proposed pharmaceuticals) thermally decompose below their boiling points and have low vapor pressures, so it is impossible to determine their normal boiling point directly, and even determining their boiling point at reduced pressure can be difficult. If the b.p. is not available in the literature, it could be quite problematic to have to purchase or synthesize the molecule to make the b.p. measurement before being able to predict its solvation free energy. Some of the other properties are even more difficult to obtain and are not available for many solvents or solutes. How do the authors propose to tackle this problem?

- The method for selecting the final models used is interesting. It would be useful to have schematics explaining this methodology.

Results and discussion:

- Figure 6, how do these compare to state of the art machine learning models that predict solvation free energies directly from molecular identifiers – so without the need for PCM calculations and macroscopic properties. If results are comparable, what is the advantage of your methodology considering the significantly higher computational time for PCM calculations compared to direct application of a machine learning model on molecular identifiers, and also the difficulties obtaining macroscopic properties discussed above?

- The results are good and show that chemical accuracy required for advanced modeling can be obtained using this methodology. However, it would be interesting to know what the application range of this method is and how it performs on new solvents or new solutes that are not present at all in the training data. Did the authors test the performance of the model on a test set of completely out-of-range data, e.g. a case where the test set included an element or functional group that was not present in the training set?

- What is the temperature range at which this method is applicable? Is it limited to 298 K or is it possible to use at higher temperatures? Does your method provide all the information needed to use the Chung et al. correlation (AIChE J. 2020; 66:e16976) to estimate the solvation energies at different temperatures?

Minor comments:

- The introduction section is long and the readability would improve by using subsections, or for example discussing the state of the art of the PCM models in a separate section.

- In the introduction, the authors talk briefly about overlooking the conformational entropy when using implicit solvents compared to explicit solvent modeling, however they do not talk about conformer sampling as a way to improve predictions. Can you make a comment about this?

- Please add an equation for RMSE on top of MUE and MUPE

- Out of the 1711 solute-solvent mixtures, how many unique solvents and solutes are considered? Please add this information.

- Can you add some more information on MRMR and how it is applied to your network? How did you choose the final input variables?

- How many layers are used in the neural network?

- The y-axis in Figure 1 is a bit confusing since the sum of the different contributions for each unsigned error do not add up to the total number of samples. Suggested improvements are plotting a fraction of the total sample on the y-axis and adding the 0.0 kcal/mol lower limit of unsigned error.

Reviewer #3 (Remarks to the Author):

The major claim of this work is that the authors have developed a machine learning (ML) neural network model that significantly reduces errors observed from widely used continuum solvation models using a variety of different model chemistries. The authors provide a C++ code that can interface with standard computational chemistry software and thus make it easily accessible to the many researchers who use of continuum solvation models. The results appear to be novel, and in principle a code that is easy to use and corrects solvation energies should be of interest to a very broad community. While I believe the manuscript is interesting work that has potential for being broadly applicable to the chemical science community, I am skeptical that this work would be appropriate for Nature Comm in its current form for several reasons.

1. The manuscript contains a relatively lengthy introduction discussing continuum solvation modeling theory dating back decades, but it does not provide much discussion about more modern solvation approaches, some of which involve ML in their parameterization schemes (10.1039/c9sc02452b, 10.1021/acs.jcim.8b00901). To provide more context for the novelty of this work, the current introduction should probably be significantly more concise so there is room for clearer justification for the authors' hypotheses for why this particular ML approach is proposed. Specifically, it might make sense to somewhat expand the explanation of different parametrization schemes for SMD, SCCS (doi: 10.1063/1.5050938), and PB (10.1063/1.5131020) solvation models. It might also help to show rising interest in ML-based computational tools for accurate predictions of chemical properties (e.g. 10.26434/chemrxiv.10052048). (Addressing this should only require minor revisions.)

2. The ML training procedures is not particularly well documented in the manuscript or the SI. ML studies should show learning curves that demonstrate the extent of over- and under-fitting with respect to the sizes of the training/validation sets (e.g. <https://machinelearningmastery.com/learning-curves-for-diagnosing-machine-learning-model-performance/>). This is not present, so it is difficult to ascertain the robustness of this model and how the different models are learning what they are learning. (Addressing this would require minor, but potentially major revisions if learning curves show problematic behaviors.)

3. Though parameterized to reaction intermediates, continuum solvation models are often used for a wide variety of chemical systems, including those from those involving ions and transition states. It is not well discussed what kinds of systems exist in the authors' training sets. Is there reason to be concerned that ML-approaches usually poorly extrapolate outside of their training sets? Would this impact the usefulness of the model in calculations involving ions and transition states? (Addressing this would probably require major revisions in the form of some additional predictions using their models relative to say, SMD calculations.)

4. The computational approach here makes use of neural networks to learn errors in solvation energies in common continuum solvation models using a variety of different model chemistries. An interesting point is how this work shows a systematic way to leverage ML in a sensitivity analysis on energy contributions from QM, and normal boiling point appears to be an important factor. I would be curious to know if other more modern solvation models that are generally considered more accurate (e.g. COSMO-RS or ESM methods) are in fact already accounting this feature explicitly or correcting for it implicitly. (Addressing this would require minor revisions.)

To summarize, I think this is intriguing work that demonstrates a straightforward model for correcting solvation energies that can interface with standard quantum chemistry codes. I think I understand the scientific hypotheses for developing this specific model and how/why it was

trained, but more concise text would help in that regard. I have concerns that this approach may not be robust for extrapolations for some ions and transition states, which would limit the use of their straightforward code, and this should be scientifically addressed. Since comments on reproducibility were requested for this report, I note that I do not see how this ML model itself could be reproduced by anyone else without scripts and training set data that would ideally be made freely available to the public via GitHub or some other repository. Perhaps these were included in the SI and I could not find them.

In closing, while this is nice and intriguing work and potentially very useful for a broad community of scientists, I think this work warrants some more considerations, and would probably be more appropriate in a more specialized chemical physics/physical chemistry journal. If the authors can concisely address the points above and show that there is no need for concern about extrapolations (as well as address other concerns raised by the editor and other reviewers) I think this work might still be appropriate for Nature Comm.

-- John A. Keith

Dear Editor, Dear Reviewers,

On behalf of myself and my co-author, I would like to thank you very much for your time and for accepting our work to review. We also appreciate the careful review and fruitful comments of the reviewers and found them very helpful in improving the quality of our work.

Before the point by point responses to the reviewers' comments, I draw your attention to the major changes that we have made in our revised manuscript:

- In our revision we describe only those of our models that require the dielectric constant of the solvent as the only experimentally determined data item, and this is needed by all continuum solvation models as well. So, the additional experimental properties used in the previous manuscript version are not required any more.
- As no additional experimentally determined data was required by the new models, we could increase our benchmark dataset from 1711 samples in the previous version of our MS to 2493 solute-solvent mixtures in our revised manuscript.
- We provide a detailed comparison of our results with most accurate models proposed in the literature, including COSMO-RS and SM12.
- For the solvation free energy predicted via COSMO-RS and the energy components of the PCM continuum solvation models, we could achieve MUE as low as 0.30208 kcal/mol which is the most accurate result ever reported for evaluation of solvation free energies of general solute-solvent mixtures.

We hope to have addressed all major concerns with our responses and revisions. We would be happy and prepared to provide further clarifications and appropriate revision if required.

Best regards,

Amin Alibakhshi

Reviewer #1

The authors develop a machine learning based continuum solvation model with an aim to improve the predictability of solvation free energy. The performance and reliability of the model are tested in comparison with other currently available models on the market. As a manuscript submitted to Nature Communications, I had expected a more advanced implicit solvation model augmented with some machine-learning-based technique or an SCRF-applicable QM machine learning model, for example, <https://doi.org/10.1126/sciadv.aav6490>, while I was reviewing the manuscript. However, I found that the proposed model is just a neural-network-based SPR model that uses several QM results and empirical properties for molecular description, similar to other published models. Nevertheless, the authors claim that their proposed model is ML-PCM.

Thank you for your comments. First of all: If defined broadly enough, essentially all models and algorithms in chemistry are just SPRs, because almost always the structures of molecules are relevant for their properties. For less broad definitions of SPRs, we maintain that what we propose is a machine-learning-based PCM model, because similar to other conventional continuum solvation models it is based on considering the solvent as a continuous medium and calculating the solvation energy components of a solute placed in the cavity of this medium by the SCRF procedure. Conventional PCM models employ simple and

ad hoc expressions to integrate and modify those calculated energy components. In contrast, we employ machine learning for this purpose and show its efficiency in substantial improvement of the predictability of solvation free energy, which is also the main novelty of our work. More importantly, as we explained in our revised manuscript, without requiring any experimentally determined data of solvent or solute, our new method allows us to achieve MUE of as low as 0.30208 Kcal/mol which is the most accurate reported result to date.

The publication cited by the reviewer is indeed interesting but also has several issues. First and foremost, in that work a neural network has been trained to reproduce solvation FE predicted by the SMD solvation model. However, as we have discussed in our manuscript, SMD itself does not provide sufficient accuracy for many solute-solvent mixtures. Another serious issue is the very large number of parameters which are used by this model (about 32k) which greatly increases the risk of overfitting. Finally, with a MUE of 1.1 kcal/mol, the accuracy which this model provides -is among the least accurate reported results (see table 2 in our revised manuscript for details).

Although the proposed model showed reasonable results, I do not see neither any novelty nor physical insight of the manuscript considering other published works, for example,

<https://doi.org/10.1039/C9SC02452B>,<https://doi.org/10.1039/C8CP07562J>,<https://doi.org/10.1063/5.0012230>,<https://10.1021/acs.jcim.0c00600>.

With the exception of <https://doi.org/10.1039/C9SC02452B>, we have discussed these other works in our previous MS version and in more detail in our revised MS. As it can be seen in table 2, all of these models suffer from being limited to only one solvent (water) as well as limited accuracy. We refrained from including <https://doi.org/10.1039/C9SC02452B> in our comparison because of the following serious issues:

- 1- This model is based on how a molecule is written as Smile string. So very little physical chemistry has been taken into account, which can make the results unreliable when applied for new and independent datasets. The very small test set also does not help in addressing this concern.
- 2- These authors have used 3 neural networks, for each compound they considered a 300-elements vector as input variable, and they devoted only 10% of the dataset to test the results. Although they have not reported specific details about the number of neurons, based on the mentioned quantities it is very likely that the ratio of samples to model parameters is far too low, resulting in a high risk of overfitting

Specific comments are as follow:

1.The authors mainly compare their results with three implicit solvation models: SMD, IEF-PCM, and C-PCM. There are more advanced and accurate models like COSMO-RS or SM12, and they also offer published results for the same database:

<https://doi.org/10.1021/ct300900e>,<https://doi.org/10.1021/jp511158y>.

In our revision, we are reporting a detailed comparison to all mentioned models as well as several further works (see table 2 for details)

2.Since the training task uses relatively small data points, the authors need to discuss the model's robustness to overfitting.

In our revision, we have used a dataset of 2493 samples and as we clearly explained in our previous version and in the revised manuscript, we have applied a very rigorous and demanding validation strategy (see the section 2 for details).

3.The proposed model uses QM-calculation results as molecular descriptors. At least the authors must account for the proposed model has distinct advantages than other SPR models, which do not require such expensive calculations for description.

As noted above, the main advantageous of the new method is its much higher accuracy, flexibility in improving performance of any solvation model and dealing with physical chemistry of the problem via working with SCRF calculated quantities rather than some ad-hoc representations.

#####

Reviewer #2 (Remarks to the Author):

Comments to “Improved prediction of solvation free energies by Machine-Learning Polarizable Continuum solvation Model”

The manuscript proposes a machine learning model to improve results from PCM calculations using quantum molecular descriptors and additional macroscopic properties. The methodology clearly improves the results obtained by PCM calculations to chemical accuracy required for modeling on a random selection of experimental data. Even though the results look good, I have some concerns related to necessity of available macroscopic properties as input to the machine learning model, as well as the application range of the model.

Thank you very much for your helpful comments. To address this issue, in our revision, we introduce models which require no additional properties other than the dielectric constant of the solvent.

Also, there is a widely used commercially-available program COSMO-RS / COSMOtherm developed many years ago which is similar (starts from a type of PCM calculation, then adds a few corrections to achieve higher accuracy), but the authors apparently haven't compared their method to COSMO. To this reviewer, it is not clear how the new method is novel compared to COSMO-RS/COSMO therm, which appears to be essentially the same concept.

As we demonstrated in our revised manuscript, our developed method allows a remarkable improvement even for the results obtained via COSMO-RS. It can also be easily used for any other solvation model in the same way. This is one of the main advantages of the proposed method. For other proposed models also although the MUE was higher than that of COSMO-RS, in terms of maximum unsigned error our method yielded better results.

More important than the novelty, the authors need to show that their empirical correction to PCM is significantly more accurate or more convenient than COSMO. (I think in reality COSMO is more convenient since it doesn't require some inputs, e.g. the normal boiling point of the solute, which is often unavailable.)

We have addressed this comment in our revision, by eliminating the need for other property inputs. And as you can see from the instructions provided as supplementary material, using the C++ code which we provided, practical use of our work is quite convenient and it can be applied on the calculated SCRF energy terms obtained by any other code.

Major Comments:

1. Introduction

- The authors claim that one of the advantages of their approach is the strong reduction in the need for macroscopic properties that are usually not readily available. The introduction should highlight more what these properties are and how the properties used in this work are more easily accessible. Some of the macroscopic properties required by the current method seem quite difficult to obtain in many practical situations.

c.f. our response to the previous comment

- With regard to importance to this work, I believe that there are 3 important topics missing in the introduction:

1. The state of the art COSMO-RS method developed by Klamt and implemented in COSMOtherm. This method typically outperforms other PCM, such as SMD. Since this software is licensed, we understand that this method is not used in the present work and the authors focus on other PCM methods for comparison, but it should at least be discussed in the introduction. We note that the authors apparently were willing to use (and implicitly to require anyone using their method to also use) the commercial GAUSSIAN software, which has a very restrictive license. Why were they unwilling to use the COSMOtherm software which is so relevant to their research? Also a comparison of the uncertainty of this method to the new ML approaches would help the manuscript to compare to the state of the art.

We have addressed this comment in our revised manuscript. About the software, we used Gaussian because our university provides a campus license for all researchers for it (but not for COSMOtherm), and more importantly, as a general purpose quantum chemistry software it is much more extensively used and has a broader range of users than COSMOtherm for which the application is mainly limited to calculating the solvation free energy. Nevertheless, it should be noted that our work is not limited only to Gaussian results. As our model inputs are the common and usual components of almost any solvation model, it can be used for these quantities computed by any other code as well.

2. The uncertainty of each of the PCM models used should be discussed (apart from the results in Table 2). How do the selected PCM models compare in general to other PCM models, do they perform better on a separate test set and is this the reason they are selected? Do they require fewer macroscopic properties compared to other methods and which macroscopic properties are required?

We have provided a detailed comparison in our revised manuscript (see table 2 for details)

3. What is the applicability range for the used PCM models? Typically these models are parameterized to perform well within a certain subset of chemical space. i.e. what is the Scope of the new method proposed in this paper?

As can be seen from the list of studied samples provided as supplementary material, our benchmark consists of a wide range of solute-solvent mixtures from a very diverse chemical families.

2. Computational details

- The geometry for the solutes is optimized in vacuo and that geometry is accounted for during computations with the implicit solvent model approach. Was there an attempt to ensure that you found the lowest-energy conformer? How do the authors account for the change in geometry of the lowest energy conformer in different solvents. Why are the geometries not optimized while accounting for the implicit solvent? Lowest energy conformers will be different in polar and apolar solvents.

Although global optimization of (conformational) energy is one of the primary research focuses of our group, for this work we refrained from searching for the lowest-energy structures as well as re-optimization for each solvent, due to the significant computational costs and complexities which this can add -- not only in developing the models but also in convenience of application for the end users. By construction and intention, implicit-solvent models are a computationally cheap, approximate substitute for large-scale, long-time explicit-solvent simulations. Therefore, a typical implicit-solvent user will not want to perform extensive additional calculations of any kind. And it was actually the task of machine learning to learn how to cope with inaccuracies resulting from such simplification steps and to correct for it.

- The authors agree that the MUPE might be a more appropriate metric to define the error for solvation free energies compared to MUE. However, many of the solvation free energies have values close to 0.0 kcal/mol, which can complicate the use of MUPE and result in near-infinite values. Although it is not common, we think it is possible that for some solute-solvent pairs the solvation free energy could be negative which would cause a problem with the MUPE formula. How do the authors handle this?

That is quite true. Accordingly, in our revision we have only reported our results with MUE and RMSE.

- Can the authors give an indication of the computational time for each of the different methods used?

It totally depends on the computational resources, level of theory, size of the molecule and applied continuum-solvation model. So it can vary from few seconds to few hours depending on the above-mentioned factors. After QM computations are finished, getting the results by the ML model is done in a fraction of a second.

- The authors use 9 macroscopic properties to optimize the neural network. At least two of those, the normal boiling point of the solute and the dielectric constant of the solvent, are very important inputs to the neural network. The authors claim that these are readily available, which is the case for common small-molecule solvents and solutes. However, when accounting for new molecules these properties are more difficult to predict and the model heavily relies on their availability. Also many larger solutes of great interest (e.g. known or proposed pharmaceuticals) thermally decompose below their boiling points and have low vapor pressures, so it is impossible to determine their normal boiling point directly, and even determining their boiling point at reduced pressure can be difficult. If the b.p. is not available in the literature, it could be quite problematic to have to purchase or synthesize the molecule to make the b.p. measurement before being able to predict its solvation free energy. Some of the other properties are even more difficult to obtain and are not available for many solvents or solutes. How do the authors propose to tackle this problem?

This is very true and a good point. To address it, in our revision we only present models which employ the dielectric constant of the solvent as the only required experimental property. Interestingly, this does not result in a drastic reduction in the accuracy of our results, i.e., the proposed models still yield satisfactory accuracy after eliminating the need for other experimental properties.

- The method for selecting the final models used is interesting. It would be useful to have schematics explaining this methodology.

As it might distract the readers from the main purpose of this work, we would rather not elaborate so much on the neural network parts. They have been extensively described already in a previous study which used the same model development strategy (<https://doi.org/10.1016/j.aca.2018.05.015>).

Results and discussion:

- Figure 6, how do these compare to state of the art machine learning models that predict solvation free energies directly from molecular identifiers – so without the need for PCM calculations and macroscopic properties. If results are comparable, what is the advantage of your methodology considering the significantly higher computational time for PCM calculations compared to direct application of a machine learning model on molecular identifiers, and also the difficulties obtaining macroscopic properties discussed above?

Yes, PCM calculations do incur substantial computational costs. These costs buy us and the users significantly better transferability to other systems, because continuum-solvation approaches take into account substantial parts of the physics behind the problem. Directly connecting molecular structures or identifiers with solvation free energies by machine learning is cheaper to use but much more difficult to develop and prone to sudden, catastrophic failures when applied to systems not within the scope of the training set, with the boundary of this scope being fuzzy or even ill-defined. This behavior is to be expected from the design of these methods. Demonstrating it without being accused of treating the ML model unfairly is very hard. – We also use ML in our method, not to bridge across all of the physics but only to post-correct it, so this is significantly more robust (cf. next question).

- The results are good and show that chemical accuracy required for advanced modeling can be obtained using this methodology. However, it would be interesting to know what the application range of this method is and how it performs on new solvents or new solutes that are not present at all in the training data. Did the authors test the performance of the model on a test set of completely out-of-range data, e.g. a case where the test set included an element or functional group that was not present in the training set?

Yes, we did. As explained in section 2, we have assigned 40% of the dataset for validation and test of the model and only 60% of the dataset has been used for training. And the results which we report are those obtained for a randomly selected and totally independent test dataset accordingly.

- What is the temperature range at which this method is applicable? Is it limited to 298 K or is it possible to use at higher temperatures? Does your method provide all the information needed to use the Chung et al. correlation (AIChE J. 2020; 66:e16976) to estimate the solvation energies at different temperatures?

Similar to the majority of implicit solvation models, our work is also mainly applicable to calculate the solvation free energy at 298K. The main reason for this is that the energy terms calculated by current solvation models, which are used as model input in our work, are commonly inaccurate at other temperatures. Nevertheless, using several thermodynamics models like the work you mentioned, it would be possible to use this solvation free energy to estimate the values at other temperatures. However, as we did not have access to a suitable benchmark, we did not elaborate on this in our work.

Minor comments:

- The introduction section is long and the readability would improve by using subsections, or for example discussing the state of the art of the PCM models in a separate section.

We considered this comment in our revision and made the introduction more concise.

- In the introduction, the authors talk briefly about overlooking the conformational entropy when using implicit solvents compared to explicit solvent modeling, however they do not talk about conformer sampling as a way to improve predictions. Can you make a comment about this?

Obviously the correct physical consideration requires conformer sampling and it is the subject of another study which we have carried out recently. Nevertheless, as we also discussed in the manuscript, in the implicit solvent approach, conformer sampling is not carried out and inaccuracies due to not properly sampling the conformations are accounted for by some ad hoc modifications. In our model, it has been taken care of by the machine learning part of the work, as mentioned in another reply above.

- Please add an equation for RMSE on top of MUE and MUPE

We added the relationship in our revised manuscript

- Out of the 1711 solute-solvent mixtures, how many unique solvents and solutes are considered? Please add this information.

In our revision we studied 2493 binary mixtures of 435 neutral solutes and 91 solvents. We added details to the first paragraph of the second section.

- Can you add some more information on MRMR and how it is applied to your network? How did you choose the final input variables?

MRMR is in fact a variable selection algorithm and allows selecting most relevant and least redundant variables from a pool of potential candidates. As elaborating on the details of the algorithm could be totally out of the scope of the present work, we refrained from doing it in our manuscript and provided a reference in the manuscript for readers who are interested to know more details.

- How many layers are used in the neural network?

Only one hidden layer, which is also declared in our revised manuscript.

- The y-axis in Figure 1 is a bit confusing since the sum of the different contributions for each unsigned error do not add up to the total number of samples. Suggested improvements are plotting a fraction of the total sample on the y-axis and adding the 0.0 kcal/mol lower limit of unsigned error.

This figure has been removed in our revision. In the previous version, it was used to show the number of compounds with unsigned error greater than specified values. Accordingly the compounds with unsigned errors lower than the first specified values were not included.

Thanks once again for your helpful comments.

Reviewer #3 (Remarks to the Author):

The major claim of this work is that the authors have developed a machine learning (ML) neural network model that significantly reduces errors observed from widely used continuum solvation models using a variety of different model chemistries. The authors provide a C++ code that can interface with standard computational chemistry software and thus make it easily accessible to the many researchers who use of continuum solvation models. The results appear to be novel, and in principle a code that is easy to use and corrects solvation energies should be of interest to a very broad community. While I believe the manuscript is interesting work that has potential for being broadly applicable to the chemical science community, I am skeptical that this work would be appropriate for Nature Comm in its current form for several reasons.

1. The manuscript contains a relatively lengthy introduction discussing continuum solvation modeling theory dating back decades, but it does not provide much discussion about more modern solvation approaches, some of which involve ML in their parameterization schemes (10.1039/c9sc02452b, 10.1021/acs.jcim.8b00901). To provide more context for the novelty of this work, the current introduction should probably be significantly more concise so there is room for clearer justification for the authors' hypotheses for why this particular ML approach is proposed. Specifically, it might make sense to somewhat expand the explanation of different parametrization schemes for SMD, SCCS (doi: 10.1063/1.5050938), and PB (10.1063/1.5131020) solvation models. It might also help to show rising interest in ML-based computational tools for accurate predictions of chemical properties (e.g. 10.26434/chemrxiv.10052048). (Addressing this should only require minor revisions.)

We tried to implement all the mentioned points in our revision.

2. The ML training procedures is not particularly well documented in the manuscript or the SI. ML studies should show learning curves that demonstrate the extent of over- and under-fitting with respect to the sizes of the training/validation sets (e.g. <https://machinelearningmastery.com/learning-curves-for-diagnosing-machine-learning-model-performance/>). This is not present, so it is difficult to ascertain the robustness of this model and how the different models are learning what they are learning. (Addressing this would require minor, but potentially major revisions if learning curves show problematic behaviors.)

In this work we used a totally different validation and training strategy based on a previous work (<https://doi.org/10.1016/j.aca.2018.05.015>). Briefly, it is similar to a 100-fold cross validation and using a t-test method to compare the test and training set results to ensure validity of our models and to reduce the risk of overfitting. Please refer to the section 2 as well as the given reference for details.

3. Though parameterized to reaction intermediates, continuum solvation models are often used for a wide variety of chemical systems, including those from those involving ions and transition states. It is not well discussed what kinds of systems exist in the authors' training sets. Is there reason to be concerned that ML-approaches usually poorly extrapolate outside of their training sets? Would this impact the usefulness of the model in calculations involving ions and transition states? (Addressing this would probably require major revisions in the form of some additional predictions using their models relative to say, SMD calculations.)

We have used 40% of our benchmark dataset for test and validation purposes, have reported our results based on what we obtained for those datasets, and have employed a t-test method to make sure both training and test sets have the same accuracy. This implies a satisfactory extrapolation of our results. As we clarified in our revision, similar to all widely accepted continuum solvation models, our model is currently only applicable to the neutral solutes, mainly due to the high inaccuracies of almost all implicit solvent approaches in evaluating the solvation free energy of ions directly, which implies significant inaccuracies in the energy components used by our work as model input. For predicting solvation free energy of ions, the implicit solvation approaches typically cannot be used directly. Instead, one can evaluate the solvation free energy using the values predicted for neutral species as well as additional data such as pKa through a thermodynamics cycle (see e.g. <https://doi.org/10.1021/jp810292n>)

4. The computational approach here makes use of neural networks to learn errors in solvation energies in common continuum solvation models using a variety of different model chemistries. An interesting point is how this work shows a systematic way to leverage ML in a sensitivity analysis on energy contributions from

QM, and normal boiling point appears to be an important factor. I would be curious to know if other more modern solvation models that are generally considered more accurate (e.g. COSMO-RS or ESM methods) are in fact already accounting this feature explicitly or correcting for it implicitly. (Addressing this would require minor revisions.)

As explained in the manuscript, COSMO-RS also proposes an ad hoc modification to correct several inaccuracy sources. Nevertheless, similar to what we carried out in our revision, it does not necessarily need to take into account boiling point explicitly, as implemented in the majority of other continuum solvation models.

To summarize, I think this is intriguing work that demonstrates a straightforward model for correcting solvation energies that can interface with standard quantum chemistry codes. I think I understand the scientific hypotheses for developing this specific model and how/why it was trained, but more concise text would help in that regard. I have concerns that this approach may not be robust for extrapolations for some ions and transition states, which would limit the use of their straightforward code, and this should scientifically addressed. Since comments on reproducibility were requested for this report, I note that I do not see how this ML model itself could be reproduced by anyone else without scripts and training set data that would ideally be made freely available to the public via GitHub or some other repository. Perhaps these were included in the SI and I could not find them.

We currently provide the code and user instructions as supporting material but we will also provide a GitHub version in the near future (as we are doing with other codes from our group, cf. e.g. <https://github.com/ogolem/ogolem>).

In closing, while this is nice and intriguing work and potentially very useful for a broad community of scientists, I think this work warrants some more considerations, and would probably be more appropriate in a more specialized chemical physics/physical chemistry journal. If the authors can concisely address the points above and show that there is no need for concern about extrapolations (as well as address other concerns raised by the editor and other reviewers) I think this work might still be appropriate for Nature Comm.

Thank you once again for your helpful comments.

Reviewer #1 (Remarks to the Author):

In the revision, following the comments of referees, the authors revised the MS substantially. The most important change is that the authors report those of the models that require the dielectric constant of the solvent as the only experimentally available data and report that the model still achieve the MUE as low as 0.3208 kcal/mol. I

I can hardly believe that this remarkable achievement is surely due to either outstanding performance of ML models or due to the increased number of benchmark data set. To make a fair comparison, the authors should have used the benchmark dataset as in the previous version of the manuscript.

Another point is that what is the rationale behind of using the dielectric constantly only other than other properties. I wonder whether the same kind of accuracy may not have been achieved if the other property (or properties) has been used.

Overall, I still do not think that the current manuscript is appropriate for a publication in *Nature Communications*. It should be more appropriate for a specific journal.

Reviewer #2 (Remarks to the Author):

First, congratulations to the authors for very significantly improving the manuscript in just a few months. This present version is suitable for publication in a journal, and one could imagine this method might become widely used (e.g. as an add-on to COSMO-RS). I am not sure if it is the right fit for publication in Nature Communications (that is a decision for the editors, not this reviewer).

I have a few technical comments.

1) Very recently my group posted a preprint demonstrating that it is indeed possible to achieve extremely high accuracy in Gsolvation estimates using ML based just on connectivity structures (e.g. SMILES), no need for 3-d geometries or quantum chemistry calculations.

<https://arxiv.org/abs/2012.11730>

That preprint has been submitted for publication in the Chemical Engineering Journal.

The model in the arxiv preprint has MUE of 0.21 kcal/mole, even better than the excellent 0.3 kcal/mole of the best models reported in the current manuscript. In our work we found that it is difficult to find test sets accurate enough to determine the accuracy of such good estimators, since many experimental Gsolvation values have errors greater than 0.3 kcal/mole. Perhaps the aleatoric errors in your test set are contributing significantly to the 0.3 kcal/mole number you report?

2) I think the authors did not understand my suggestion about using a test set that is chemically distinct from the training set. Random splits usually give test sets that have similar molecules, functional groups, and elements as those in the training set. This is good for testing accuracy within that chemical space, but does not tell you much about how well the estimator will work on completely new chemistry. To better test how well the model extrapolates to new chemistry, you can use splits where certain solvents or solutes or functional groups were excluded entirely from the training set. Or certain elements were excluded entirely. The arxiv preprint mentioned above includes some examples testing the extrapolative power, take a look.

3) There are many different versions of COSMO-RS. The corrections in each version of the commercial COSMO-RS model are designed to be used with specific functionals and basis sets. For example, the most recent COSMO-RS model is supposed to be used with a specific TZVP-Fine basis set available in TURBOMOLE (but not in GAUSSIAN), and with a certain DFT functional. Even the auxiliary basis set employed matters (and is specified). But in the manuscript, you seem to be using two different DFT methods and some different basis sets (and probably auxiliary basis sets) than what modern COSMO-RS versions expect. I think this is OK, but it should be noted in the text, and you should carefully state exactly how you did the calculations and which version of COSMO-RS you used.

4) I am not sure if it is completely fair to compare your model to the 2015 version of COSMO-RS reported in Ref. 49 as you did in your table. You have the COSMO-RS numbers you computed using your basis sets and functionals; do they also give similar MUE as Klamt found in 2015 using his basis sets, functionals, and version of COSMO-RS at that time?

5) One would think that the distributors of COSMO-RS would like to include your ML model to improve their estimates, and certainly their customers would like to be able to predict more accurate solvation energies. I suggest you have a conversation with them about this, it could be a great way for your work to have real-world impact. I appreciate that the readers of your paper could take your software, and buy COSMO-RS separately, and then follow your method to make improved calculations. But it would be much more convenient if your model were included e.g. as an option in the widely-distributed version of COSMO-RS.

Reviewer #3 (Remarks to the Author):

The authors have satisfactorily addressed my comments, and I believe they have also addressed comments made by other reviewers. I think the potential impact of this work is improved and better described and therefore I believe this would be suitable for publication in Nature Commun.

Dear Reviewers,

On behalf of myself and my co-author, I would like to thank you very much for considering our work for further revisions.

Before the point by point responses to the reviewers' comments, I draw your attention to the major changes that we have made in our revised manuscript:

- In our revision, we employed an alternative training strategy motivated by the comments of reviewer #2. Accordingly, for all models, the validation and test sets were always selected from samples with either a solvent or solute element not employed in the training set.
- Using the training strategy mentioned above, which allowed us to improve transferability of our models, we decided to study larger sizes of neural networks with up to 140 neurons in the hidden layer. As a result, we could reduce the obtained MUE to 0.24387 kcal/mol.
- We provided the performance of the develop models in out-of-sample tests and could show reliability of our developed models for out-of-sample prediction of solvation free energy.

Similar to the previous submission, we have provided two versions of our revised manuscript depending of having the revisions highlighted (“manuscript_highlighted”) or not (“manuscript_clean”).

We hope to have addressed all major concerns with our responses and revisions. We would be happy and prepared to provide further clarifications and appropriate revision if required.

Best regards,

Amin Alibakhshi

Point by point responses to Reviewer #1,

Comment: In the revision, following the comments of referees, the authors revised the MS substantially. The most important change is that the authors report those of the models that require the dielectric constant of the solvent as the only experimentally available data and report that the model still achieve the MUE as low as 0.3208 kcal/mol. I can hardly believe that this remarkable achievement is surely due to either outstanding performance of ML models or due to the increased number of benchmark data set.

Response: Thank you for your comments. We would like to clarify here that the MUE of 0.3208 is not obtained for the very same models that we introduced in the first version. They are for new models which take COSMO-RS free energies as additional model inputs and this has resulted in remarkable improvement. For the same initially proposed models, as we declared in the manuscript, by limiting the input variables to only the dielectric constant, the MUE as been increased to 0.53029 kcal/mol.

Comment: To make a fair comparison, the authors should have used the benchmark dataset as in the previous version of the manuscript.

Response: Generally, the larger the size of the dataset used to train a model, the more reliable that model is. Nevertheless, to address this comment, we provided details on the performance of the models over individual samples, stated as maximum unsigned error, and also the graphs depicting the comparison of the reference and predicted data and through them we could demonstrate that the results are not affected by increasing the system size.

Comment: Another point is that what is the rationale behind of using the dielectric constantly only other than other properties. I wonder whether the same kind of accuracy may not have been achieved if the other property (or properties) has been used.

Response: We made this decision according to the comments of Reviewer#2. Please refer to his first comment and our given responses as well as to the first part of our general response to reviewer comments.

Reviewer #2 (Remarks to the Author):

First, congratulations to the authors for very significantly improving the manuscript in just a few months. This present version is suitable for publication in a journal, and one could imagine this method might become widely used (e.g. as an add-on to COSMO-RS). I am not sure if it is the right fit for publication in Nature Communications (that is a decision for the editors, not this reviewer).

I have a few technical comments.

1) Very recently my group posted a preprint demonstrating that it is indeed possible to achieve extremely high accuracy in Gsolvation estimates using ML based just on connectivity structures (e.g. SMILES), no need for 3-d geometries or quantum chemistry calculations.

<https://arxiv.org/abs/2012.11730>

That preprint has been submitted for publication in the Chemical Engineering Journal.

The model in the arxiv preprint has MUE of 0.21 kcal/mole, even better than the excellent 0.3 kcal/mole of the best models reported in the current manuscript. In our work we found that it is difficult to find test sets accurate enough to determine the accuracy of such good estimators, since many experimental Gsolvation values have errors greater than 0.3 kcal/mole. Perhaps the

aleatoric errors in your test set are contributing significantly to the 0.3 kcal/mole number you report?

Response:

Many thanks for your comments and also congratulations for your very interesting work which was very inspiring for us in improving our work. Yes, that's quite true. We have mentioned and discussed this limitation in our revision (last paragraph in page 10).

2) I think the authors did not understand my suggestion about using a test set that is chemically distinct from the training set. Random splits usually give test sets that have similar molecules, functional groups, and elements as those in the training set. This is good for testing accuracy within that chemical space, but does not tell you much about how well the estimator will work on completely new chemistry. To better test how well the model extrapolates to new chemistry, you can use splits where certain solvents or solutes or functional groups were excluded entirely from the training set. Or certain elements were excluded entirely. The arxiv preprint mentioned above includes some examples testing the extrapolative power, take a look.

Response:

Thank you for the very interesting suggestion. We have used this idea both to examine the performance of our models in out-of-sample predictions and also to revise our training strategy exploiting solvent or element splits between training, validation and test sets, as discussed now in the revised manuscript (see tables 2 and 3 in page 9).

3) There are many different versions of COSMO-RS. The corrections in each version of the commercial COSMO-RS model are designed to be used with specific functionals and basis sets. For example, the most recent COSMO-RS model is supposed to be used with a specific TZVP-Fine basis set available in TURBOMOLE (but not in GAUSSIAN), and with a certain DFT functional. Even the auxiliary basis set employed matters (and is specified). But in the manuscript, you seem to be using two different DFT methods and some different basis sets (and probably auxiliary basis sets) than what modern COSMO-RS versions expect. I think this is OK, but it should be noted in the text, and you should carefully state exactly how you did the calculations and which version of COSMO-RS you used.

Response:

We actually didn't recalculate the COSMO-RS free energies as we do not have access to the COSMOtherm software. We only used the reported free energies which were calculated with this method in the provided reference and used it as an additional model input. We would also believe if we can use the more recent versions of the COSMO-RS and also use the energy terms computed with this method, it will probably result in more accurate predictions of the solvation free energy. However, as we currently do not have access to that software, we could not examine it. We clarified it, however, in our revision.

4) I am not sure if it is completely fair to compare your model to the 2015 version of COSMO-RS

reported in Ref. 49 as you did in your table. You have the COSMO-RS numbers you computed using your basis sets and functionals; do they also give similar MUE as Klamt found in 2015 using his basis sets, functionals, and version of COSMO-RS at that time?

Response:

As mentioned in response to the previous comment, they are computed using the calculated free energies reported in the provided reference and we did not recalculate them.

5) One would think that the distributors of COSMO-RS would like to include your ML model to improve their estimates, and certainly their customers would like to be able to predict more accurate solvation energies. I suggest you have a conversation with them about this, it could be a great way for your work to have real-world impact. I appreciate that the readers of your paper could take your software, and buy COSMO-RS separately, and then follow your method to make improved calculations. But it would be much more convenient if your model were included e.g. as an option in the widely-distributed version of COSMO-RS.

Response:

That is also a very good suggestion. Thank you. It is for sure our plan to improve the current results and also discuss them with the developers of COSMO-RS in the near future.

Reviewer #3 (Remarks to the Author):

The authors have satisfactorily addressed my comments, and I believe they have also addressed comments made by other reviewers. I think the potential impact of this work is improved and better described and therefore I believe this would be suitable for publication in Nature Commun.

Response:

Thanks so much.

Reviewer #2 (Remarks to the Author):

The authors have done a nice job revising the manuscript. I believe this version is suitable for publication.

Dear Editor, Dear Reviewers,

On behalf of myself and my co-author, I would like to thank you very much for accepting our work for publication. We also thank the reviewers for their great comments and careful review. It was for sure very helpful in improving the quality of our work.

Best regards,

Amin Alibakhshi

Reviewer #2 (Remarks to the Author):

The authors have done a nice job revising the manuscript. I believe this version is suitable for publication.

Response:

Thanks so much for all your comments and also finding our work suitable for publication.